# Residential Proximity to Urban Play Spaces and Childhood Overweight and Obesity in Barcelona, Spain: A Population-Based Longitudinal Study

**DOI:** 10.3390/ijerph192013676

**Published:** 2022-10-21

**Authors:** Nacho Sánchez-Valdivia, Carmen Pérez-del-Pulgar, Jeroen de Bont, Isabelle Anguelovski, Antonio López-Gay, Andrea Pistillo, Margarita Triguero-Mas, Talita Duarte-Salles

**Affiliations:** 1Barcelona Lab for Urban Environmental Justice and Sustainability, Institute of Environmental Science and Technology (ICTA), Universitat Autònoma de Barcelona (UAB), 08003 Barcelona, Spain; 2Hospital del Mar Medical Research Institute (IMIM), Carrer Doctor Aiguader, 88, 08003 Barcelona, Spain; 3Helmholtz Centre for Environmental Research—UFZ Department Environmental Politics, 04318 Leipzig, Germany; 4Department for Political Science, Friedrich-Schiller-University, 07737 Jena, Germany; 5Institute of Environmental Medicine, Karolinska Institute, 171 77 Stockholm, Sweden; 6ICREA (Institució Catalana de Recerca i Estudis Avançats), 08010 Barcelona, Spain; 7Department of Geography, Universitat Autònoma de Barcelona (UAB), 08193 Barcelona, Spain; 8Center for Demographic Studies, 08193 Bellaterra, Spain; 9Fundació Institut Universitari per a la Recerca a l’Atenció Primària de Salut Jordi Gol i Gurina (IDIAPJGol), 08007 Barcelona, Spain; 10Mariana Arcaya’s Research Lab, Department of Urban Studies and Planning, Massachusetts Institute of Technology, 77 Massachusetts Ave, Cambridge, MA 02139, USA

**Keywords:** urban health, built environment, childhood, overweight, obesity, body mass index, play space, playground, park

## Abstract

Findings on the relationship between play spaces and childhood overweight and obesity are mixed and scarce. This study aimed to investigate the associations between residential proximity to play spaces and the risk of childhood overweight or obesity and potential effect modifiers. This longitudinal study included children living in the city of Barcelona identified in an electronic primary healthcare record database between 2011 and 2018 (N = 75,608). Overweight and obesity were defined according to the WHO standards and we used 300 m network buffers to assess residential proximity to play spaces. We calculated the risk of developing overweight or obesity using Cox proportional hazard models. A share of 29.4% of the study population developed overweight or obesity, but we did not find consistent associations between play space indicators and overweight or obesity. We did not find any consistent sign of effect modification by sex, and only some indications of the modifying role of area socioeconomic status and level of exposure. Although it is not possible to draw clear conclusions from our study, we call for cities to continue increasing and improving urban play spaces with an equitable, healthy, and child-friendly perspective.

## 1. Introduction

Levels of childhood overweight and obesity are on the rise worldwide, particularly in Global South countries and urban contexts [1]. In 2020, 5.7% of children under 5 years old in the world were living with overweight [2]. In Spain, although the overall trend of prevalence and incidence of childhood overweight and obesity between 2005 and 2017 has slightly decreased [3], rates are still among the highest in Europe [4]. Indeed, increasing inequalities have been observed over time in deprived areas and non-Spanish communities [5]. Childhood overweight and obesity is associated with developing non-communicable diseases and psychosocial problems during childhood and adulthood [6,7,8,9].

Play spaces (i.e., playgrounds, plazas, parks, green play spaces, urban forest recreational areas, and sports fields) have been recognized as an urban built environment component promoting health and wellbeing among children. The role of these spaces in, for example, enhancing physical activity and reducing obesity, contributing to children development, and promoting good mental health has been widely illustrated [10,11,12,13,14,15]. In recent decades, local governments have shown interest in creating outdoor play spaces [16,17,18] and promoting more playful cities for children, including in cities such as Barcelona, Nantes, or Amsterdam. In this paper, we would like to offer new evidence that could support the planning of healthy and child-friendly cities, and respond to calls from organizations such as UNICEF [19] and UN-Habitat that globally defend “the right of the child to play” through programs such as the Child Friendly Cities initiative.

Previous research on the relationship between play spaces and childhood overweight and obesity has reported some mixed results, probably due to differences in exposure and outcome measure, study design, and context singularities, since most of them were conducted in North American’ settings where urban form, density, and design greatly varies. Several studies have shown that availability of green play spaces (specifically parks and urban forests) and playgrounds are related to lower rates of overweight and obesity [15,20,21,22,23,24,25,26]. Similarly, prenatal and early life exposures to green spaces have been associated with small reductions in body mass index (BMI) [27]. However, a small number of studies in the United States, Canada, and Germany have found null associations between play spaces (specifically parks and playgrounds) and childhood overweight and obesity among preschool aged children (3–5 years old) and young children (6–11 years old) [28,29,30,31,32].

Some research—mostly conducted once again in North America—has examined existing demographic and socioeconomic inequalities in the relationship between play spaces and childhood overweight [15,21,24,26,30]. These studies have identified disparities by individual conditions, such as age and gender (measured using sex). Some have shown that boys might benefit more from living close to green spaces and parks than girls [15,26], although others have found that the number of parks and playgrounds were associated with lower body mass index for girls’ exposure [21,24]. A beneficial effect of residential play spaces availability on BMI has been found for older children (10 and 13 years old) in some studies [15,21]. Race and ethnicity also may have a moderating role in the play space–overweight relationship [21,24,33]. For example, when compared with white girls, African American girls’ BMI has been associated with girls benefiting from higher residential playground availability [24]. Differences have also been observed by the intersection of family socioeconomic status (SES) with sex and race [24,33]. In the case of boys with low SES, play spaces have been associated with lower BMI, whereas for girls the relationship has been found to be in the opposite direction [24].

Overall, children living in disadvantaged neighborhoods may be excluded from beneficial environmental conditions, as they may have more difficulty accessing these spaces or feeling welcomed in them [19,34]. Nevertheless, little research exploring changes in the association between play spaces and overweight by areas’ socioeconomic characteristics has been conducted in Europe, where cities tend to be denser and have a greater number of neighborhood play places [29,35,36]. The effect of play spaces’ features, such as quality, on overweight and obesity, has been less studied [37,38]. For instance, the diversity of playing areas and the presence of green spaces have been outlined as important factors benefitting children’s health [14,27,39]. 

Given the above, the objective of this study is to investigate whether residential proximity to play spaces is associated with overweight or obesity risk in children between 2 and 14 years old in Barcelona, to assess if there are unequal effects by sex and area socioeconomic characteristics and level of exposure to play spaces. In addition, we explore the role of multiple urban play space indicators (overall, green, and diversity) in relation to overweight and obesity. In this sense, our study aim was to better explore these relationships through a longitudinal study with a large cohort of children (2011–2018), since most previous evidence has been based on cross-sectional studies, and to assess the exposure to more than 1650 play spaces and different types of play spaces (e.g., playgrounds, plazas, recreational sports fields, parks, gardens, and urban forests) in the city of Barcelona.

## 2. Methods

### 2.1. Design and Study Population

We conducted a retrospective longitudinal open cohort study including children living in the municipality of Barcelona (Spain) between 2011 and 2018, who were part of the Information System for Research in Primary Care (SIDIAP). SIDIAP is a large pseudo-identified electronic health records database collected by health professionals during primary care visits in centers managed by the public Catalan Health Institute. It includes nearly 6 million people, which represents 80% of the population living in the region [40]. This study used data from children from the 53 primary care centers publicly managed in Barcelona (out of 67 primary care centers that exist in the city).

The municipality of Barcelona has a total of 1.6 million inhabitants, of which 10% (169,715) are children aged between 2–14 years old [41]. Barcelona has one of the highest population densities in Europe [42] and is the urban center of a metropolitan region of almost 5 million inhabitants in Catalonia (Spain). Barcelona public spaces, including plazas, playgrounds, and parks, are the main places of recreation and socialization for children [13]. In 2014, Barcelona had over 28% of its urban area (580.6 hectares) dedicated to parks [43].

We included children aged between 2 and 5 years (both included), who were identified as normal weight at baseline (first body mass index (BMI) measure) [44] and were part of Barcelona’s city SIDIAP database between 1 January 2011 and 31 December 2018. Children with at least two BMI measures within the study period and with a minimum of 1.5 years of difference between the first and last measure were included. Those with only one measure between 2011 and 2018 were included, with the assessment of the initial weight measure based on data from 2006–2010 (both years included) if available. All included individuals were followed up until they reached 15 years old, became overweight or obese, transferred-out of SIDIAP, or died, or until the end of the study period (31 December 2018) (Appendix A). The Clinical Research Ethics Committee of the IDIAPJGol approved this study (CEI 21/006-P). 

### 2.2. Exposure Assessment: Residential Proximity to Play Spaces

To measure the residential proximity to play spaces, following Pérez-del-Pulgar et al. [14], we used the number of play spaces whose surrounding 300 m network buffer intersected study participants’ residential census tract at baseline. The network buffer was built from the central point of each of the play spaces, or the main entrance in the case of fenced parks and urban forests [14]. The selection of 300 m for the buffer was defined according to the average independent walking mobility standard for children defined by UNICEF [45]. We used data on public outdoor play spaces (including playgrounds, plazas, parks, gardens, urban forests, and recreational sports fields) in the municipality of Barcelona from 2014, obtained from the Ecology and Urbanism Department. Accordingly, we included a total of 1665 play spaces in our study.

Based on these assessments on the residential proximity to play spaces, we developed three indicators: (i) overall play spaces, defined as the total number of play spaces whose 300 m network buffer intersects participants’ residential census tract; (ii) green play spaces, total number of green play spaces (i.e., parks, gardens, urban forests, and playgrounds and recreational sports fields located inside parks) whose 300 m network buffer intersects participants’ residential census tract; and (iii) diversity of play spaces, assessing the diversity of the play space 300 m network buffers intersecting the participants’ residential census tract. To calculate the diversity of play spaces, we used a Shannon index of diversity, an index commonly used in ecology, according to four types of play spaces (green play spaces (defined above), playgrounds, sport-oriented play spaces (i.e., recreational sports fields), and community public spaces (i.e., plazas)) [46]. The diversity of play spaces (Shannon index) was expressed in a logarithmic scale. Higher values of the index (i.e., closer to 1.3, the maximum range in our sample) indicate a higher quantity of types of play spaces (variety of play types offered by the proximate play spaces) and higher relative number of each typology (abundance of play spaces). Conversely, lower values (i.e., closer to 0) signal lower variety and abundance of play spaces. To estimate the indicators of residential proximity to play spaces we used ArcGIS v.10.7.

### 2.3. Outcome Assessment: Childhood Overweight and Obesity

The body height and weight, routinely measured in primary care centers, was used to calculate BMI (weight in kilograms divided by height in meters squared). BMI z-scores were calculated (zBMI, in SD units), for the specific ages and sexes using the Worldwide Health Organization (WHO) children standards and references [47,48]. On the basis of z-scores, children were categorized as overweight/obesity or normal weight. We considered as overweight/obesity those children younger than 5 years old with a zBMI equal or higher than +2 standard deviations (SD) and those children aged 5 and older with a zBMI equal or higher than +1 SD [47,48]. Children below overweight/obesity values were assigned to the normal weight group. 

### 2.4. Covariates

We selected our confounding and effect-modifying variables based on the literature [14,15,23,27,29,33,37] and based on an environmental and health justice approach, which considers the complex interplay between urban social vulnerabilities and urban interventions as greening [49,50] (Appendix A). Accordingly, from SIDIAP we obtained individual level data including sex, age at baseline, time of follow-up, and nationality and area socioeconomic status (SES). We used nationality as a proxy of migration status, and thus social vulnerability (as used in other studies in Barcelona [51]), and we defined it as a dichotomous variable with two categories: Global South (i.e., Latin America and the Caribbean, Africa, and Asia/Middle East) or Global North (i.e., Europe and Anglo-Saxon America) [13].

Area SES was assessed at the census tract level using a deprivation index [52], which considers several indicators related to work (manual and temporary workers, and unemployment), insufficient education (overall and among youth) and dwellings without Internet, from the 2011 Spanish national census. Based on the distribution of our study population, deprivation level was categorized into quintiles, in which the first quintile were the least deprived areas, and the fifth quintile, the most deprived areas. From the municipal population census we also estimated the number of children aged 2–14 living in each of the census tract areas in 2014. We used this covariate to control for the potential pressure of use of each outdoor play space, hypothesizing that play space congestion can act as a barrier for its use and diminish children’s associated health benefits [14]. 

### 2.5. Statistical Analyses

We estimated Cox hazard models following previous similar studies [53]. Accordingly, hazard ratios (HR) and 95% confidence intervals (95% CI) were estimated to quantify the magnitude of the association between the risk of developing overweight or obesity and residential proximity to play spaces (each of the three indicators in separated models). We also fitted crude and adjusted models by individual covariates (sex, age at baseline, and child nationality) and area census tract variables (area SES level and the number of children in each area). 

We analyzed the three play space indicators in two different ways: either as continuous variables per interquartile range (IQR) increase, and as categorical variables by tertiles (low, moderate, and high) of exposure, since some of the relationship deviated from linearity (Appendix A). To create the categorical variables, we calculated tertiles of exposure for each indicator and classified them as: low for the first tertile (<9 units of overall play spaces; <5 units of green play spaces; <0.85 diversity index), moderate for the second tertile (9–15 units of overall play spaces; 5–8 units of green play spaces; 0.85–0.98 diversity index), and high for the third tertile (≥15 units of overall play spaces; ≥8 units of green play spaces; ≥0.98 diversity index) of exposure. 

The linearity of the three exposures (overall, green, and diversity of play spaces) with the outcome was evaluated using generalized additive models (GAMs) (Appendix A). The presence of multicollinearity in the model was assessed using the Variance Inflation Factor (VIF). Correlations between the three different play space variables (overall, green, and diversity) were also assessed (Appendix A). Stratification was conducted for sex and area deprivation level (in quintiles) as these were the only factors that showed statistically significant interactions with health outcome results. Statistical significance was set at *p*-value < 0.05, with all tests 2-tailed. Analyses were conducted in R software (3.6.2 version, R Core Team, Vienna, Austria). 

Sensitivity analyses were performed to assess the robustness of our results. We evaluated differences regarding our main results considering only children who had developed obesity during follow-up (excluding overweight). We considered obesity in those children younger than 5 years old with a zBMI equal or higher than +3 SD and those children aged 5 and older with a zBMI equal or higher than +2 [47,48]. We also evaluated if the main effect estimates varied considering only children who did not change residency during study period. Similarly, we checked if reclassifying individual children’s nationality as a variable categorized within Spanish and non-Spanish had an effect. Finally, for those with maternal information available, we evaluated maternal nationality and maternal BMI as potential confounders [53].

## 3. Results

### 3.1. Descriptive Statistics

We included 75,608 children living in Barcelona who had normal weight at study entry (Appendix A). The distribution of the study population by assigned sex was homogenous (with 51.3% being boys) and only 6.6% had Global South nationalities (Table 1). A share of 50% had 12 overall play spaces at 300 m from their residence, of which half were green play spaces. Children were exposed to an index of diversity of outdoor play spaces below one (0.9), which means high richness and abundance of play spaces in Barcelona, because the median is close to the maximum value in our sample (1.3). Nearly one-third of children (29.4%) developed overweight or obesity between 2011 and 2018, with a median age of 6.3 years (IQR 2.2) at diagnosis (i.e., when BMI was measured). Compared to those who remained normal weight during follow-up, we found that children who developed overweight or obesity were more likely to be boys (53.6% vs. 50.3%), from the Global South (7.1% vs. 6.4%), and living in the more deprived areas (for example, in the fifth quintile of deprivation: 23.4% vs. 18.6%) (Table 1; Appendix A for sex-stratified descriptive statistics). However, we did not find differences in the distribution of residential proximity to play spaces across weight status. 

In the stratified sample by quintiles of deprivation, we found a higher proportion of children with overweight or obesity as area deprivation increases (first quintile: 23.7%; fifth quintile 34.4%) (Table 2; Appendix A). Those children living in the least deprived areas (first quintile) and in the most deprived areas (fifth quintile) had lower residential exposure to overall and green play spaces. In contrast, differences in the diversity index were small (Table 2).

To avoid collinearity in our models, we performed the analyses for each of the three play space indicators separately, as proximity to overall play spaces was strongly correlated with green play spaces (Spearman’s correlation coefficient r = 0.83) (Appendix A). 

### 3.2. Main Results

In our Cox models, we did not find a consistent statistically significant association between play spaces—either continuous or by tertile exposure (Table 3)—and development of overweight or obesity for any of the three exposures (overall play spaces, green play spaces, and diversity). We did not observe important differences between crude (not shown in this paper) and adjusted models. More specifically, in the adjusted models for continuous exposure to play spaces, each IQR (9) increase in proximate overall play spaces represents a non-statistically significant 2% risk increase in developing overweight or obesity (HR = 1.02 [95% CI 1.00; 1.03], *p*-value = 0.06). Regarding green play spaces, each IQR (5) increase was associated with a nearly statistically significant growth of 1% in the risk of developing overweight or obesity (HR = 1.01 [95% CI 1.00; 1.03], *p*-value 0.05). Moreover, a nearly statistically significant 2% risk increase in developing the disease was observed for each gain of 0.29 units of the Shannon index of diversity within 300 m from children’s census tracts (HR = 1.02 [95% CI 1.00; 1.04; *p*-value 0.05]).

When exposures were categorized by tertiles (Table 3), children exposed to the highest levels (tertile 3, T3) of overall play spaces, green play spaces, and diversity play spaces were 2% (HR = 1.02 [95% CI 0.99; 1.05]), 3% (HR = 1.03 [95% CI 0.99; 1.06]), and 3% (HR = 1.03 [95% CI 0.99; 1.07]), respectively, more likely to develop overweight or obesity compared with children exposed to the lowest levels (tertile 1, T1), although non-statistically significant related. When considering children exposed to moderate levels of play spaces (tertile 2, T2), we did not find a statistically significant association, except for the diversity indicator, for which we observed a 4% risk increase in developing overweight or obesity in adjusted models (HR = 1.04 [95% CI 1.01; 1.07]). In the sex-stratified Cox models, we did not find relevant differences between boys and girls in the relationships between play spaces and overweight or obesity (Table 3).

When stratification was performed by area deprivation level (Table 4), most results indicated that there were not consistent associations between play spaces and overweight or obesity, and the few associations we found did not present any clear pattern. For the overall play space models, when exploring them as continuous variables, we only observed a 6% increased risk of developing overweight or obesity for children living in areas in the second quintile of deprivation (the second least deprived) (HR = 1.06 [95% CI 1.02; 1.11]). We also found that children living in census tracts with high levels of overall play space exposure (tertile 3, T3) and areas from the second quintile of deprivation (the second least deprived), had a higher risk to develop overweight or obesity compared with those with a low level of exposure (tertile 1, T1) (HR = 1.09 [95% CI 1.01; 1.18]). In contrast, children living in the third least deprived areas (third quintile) and with moderate levels of exposure (tertile 2, T2) to overall play spaces had reduced risk of overweight or obesity when compared to those with low exposure (tertile 1, T1) (HR = 0.91 [95% CI 0.84; 0.98]). For residential proximity to green play spaces, moderate levels of exposure were significantly associated with reduced risk of developing overweight or obesity for children living in the least deprived areas (first quintile) (HR = 0.91 [95% CI 0.84; 0.98]) when compared to children with low levels of exposure (Table 4). When exploring diversity of play spaces near the residence, those exposed to moderate levels and living in the third least deprived areas (third quintile), the risk of developing overweight or obesity was observed to increase by 10% for each gain of 0.29 units of the Shannon index of diversity within 300 m from children’s census tracts (HR = 1.10 [95% CI 1.02, 1.18]) (Table 4).

Finally, the results of the sensitivity analyses were coherent with those of the main analyses. Non-statistically significant HRs were obtained when we evaluated the association between play spaces and the risk of developing only obesity (not including overweight as outcome) among children who were normal or overweight (HR = 1.01 [95% CI 0.98; 1.03]) (Appendix A). When the dataset was restricted to children who did not change residency during the study period, estimated effects were in line with those for the entire population (Appendix A). Moreover, the results did not change significantly after categorizing children’s nationality as Spanish or Non-Spanish, instead of using the Global North and Global South division (Appendix A), and remained similar when the models were additionally adjusted with maternal information (nationality and BMI) (Appendix A). Sensitivity analyses also showed no differential effects by sex (Appendix A). 

## 4. Discussion

### 4.1. Inconsistent Relation between Residential Proximity to Place Spaces and Overweight and Obesity

In this large cohort study, residential proximity to play spaces was not associated with overweight or obesity for children living in Barcelona. These results were also null when exploring differences by sex. However, we found some small indications of a potential modifying role of area SES and residential exposure level.

The overall results of residential proximity to play spaces did not explain differences in developing overweight or obesity. This finding contradicts our initial hypothesis (more play spaces, less risk to develop overweight or obesity, especially in deprived areas) but adds to the existing mixed evidence and raised some interesting discussion points [21,23,26,30,37,54]. Our results might be explained by different reasons concerning exposure assessment or due to residual confounding linked to household SES, and the urban singularities of the city of Barcelona and its play space distribution. Barcelona is a city with a high number of neighborhood playground spaces. However, many of those are small, overused, and dense, and provide little opportunity for active use and play. Thus, they are not as protective against overweight or obesity. Many are also located in gentrifying and high-tourism neighborhoods. This may undermine safety, access, and benefits for children, especially for those in lower-income or middle-income areas [34].

Differences in measuring the exposure to play spaces (e.g., average distance to play spaces from home, total number of play spaces near home, proportion of play spaces areas in the neighborhood, among others) [29,55] and the health outcome (e.g., overweight, obesity, increases in BMI, increases in weight) might account for the inconsistency in results in the existing literature. In our study and Potestio et al. [28], the number of different types of play spaces in children’s census tract is used, with no general consistent associations found. In contrast, some others have used the percentage of the buffer around the home that is covered specifically by green spaces, parks, or playgrounds, and have shown beneficial associations between residential play spaces and overweight or obesity [15,27,56]. A recent study performed in the city of València (Spain) concluded that park land area was related to reductions in youth BMI percentile [54]. Our study assessed the exposure to play spaces using a 300 m buffer, since it is a reasonable distance that children can travel autonomously by non-motorized means of transport (e.g., walking or cycling), and because it allowed us to measure proximity to play spaces in a dense city with abundant playing areas such as Barcelona. Indeed, there is international consensus to measure children’s daily space in the urban context using the 300 m distance established by UNICEF [45].

Furthermore, we found non-differential effects by children’s sex, which seems inconsistent with the previous evidence [15,24,26]. Actually, a study conducted by Morgan Hughey et al. [24] showed that sex, race/ethnicity, and SES moderated the associations between play spaces and weight. The lack of differential impacts by sex group in our study could be explained by our inclusion of a broad diversity of types of play spaces (i.e., playgrounds, plazas, recreational sports fields, parks, green play spaces, and urban forests). Other studies that found differences by sex were limited in terms of what they included in their exposure sample, assessing exposure considering only parks and playgrounds, which are spaces that are well-known to be more used by boys than girls [24]. 

Although our results are not consistent, we found some indications of a potential role of area SES on moderating the associations between childhood overweight or obesity and residential exposure to play spaces. On one hand, we observed a detrimental association between children with higher residential proximity to play spaces and living in less deprived areas (second quintile), which builds on previous research that related higher park availability with higher BMI for high-SES families [24]. This differential effect may be explained by the lowest dependence on, and time spent in, public space (i.e., parks and playground) for those living in wealthy areas. This is because many high-income families have more access to private green and play spaces, or can more easily access private green (and blue) spaces in their second homes, a phenomenon largely present in Spain since the 1970s and which has increased in the last twenty years [13,57]. Additionally, Barcelona is also a city in which the highest levels of air and noise contamination are experienced in medium- to high-income areas (such as the Eixample, a wealthy neighborhood, which itself has one of the poorest accesses to green space in the city) [58,59]. 

On the other hand, we found a small protective association for children with moderate exposure to play spaces but living in the least deprived areas (first quintile) (in the case of green play spaces), and also in middle deprived areas (third quintile) (in the case of overall play spaces). The green benefit for least deprived areas is in line with the idea that wealthy groups may benefit more from the positive effect of living in proximity to green play spaces in terms of health [50]. However, this positive effect differs from previous studies in the United States [21,24], which found that children with low SES were more likely to benefit from higher park and playground availability.

Related to the diversity of play spaces indicator, our study showed few adverse associations with developing childhood overweight or obesity. Although we are unaware of previous studies exploring this relation, our results are contrary to previous research assessing children’s wellbeing and other health outcomes (e.g., mental health), which emphasized a positive effect [14,60]. A recent study in Barcelona found that lower prevalence of disorders of psychological development was consistently associated with a greater diversity of play opportunities [14].

In addition to socioeconomic factors, current evidence shows that play space features might be more important in explaining the relationship between play spaces and overweight or obesity rather than number of play spaces, as we consider in our study [12,23,25,30,37,38,39,61,62,63]. For instance, parks are more likely to encourage physical activity if they are perceived as being aesthetically pleasing and having more diverse non-play and play infrastructure (minor traffic, sidewalks, trees, retail shops) [39], or if they are safe both from an infrastructure and crime standpoint [34,64]. Moreover, the quality and design of the play spaces motivate their use by children, which might help to explain the relationship between these facilities and overweight or obesity in the city of Barcelona [65]. For example, previous research shows that low-quality park amenities in highly and moderately deprived areas are associated with obesity prevalence [37]. A recent study performed in Barcelona for adults concluded that urban green quality, especially bird biodiversity and amenities, reduced the risk of overweight or obesity [38]. Accessibility is another important element to consider, since children in neighborhoods with good access to play spaces are more active and less likely to be overweight or obese [20].

Accessible, welcoming, and opened play spaces have also been found to contribute to overall children’s relational wellbeing in Barcelona through the creation of informal care networks around children and children’s positive appropriation of the space [13]. Actually, considering children’s and caregivers needs in relation with parks and playgrounds, and involving the community in their planning, led to an increasing use of play spaces and physical engagement in children [65]. That said, these characteristics were not available in our dataset.

### 4.2. A Need for a Better Understanding of Pathways

In summary, the relation between play spaces and childhood overweight or obesity is complex, and the results of our study need to be interpreted cautiously. There are several pathways that may underlay the effects of play spaces on children’s health, such as increased levels of physical activity, building social capacities, relational wellbeing, stress mitigation and restoration, and reducing harm from exposure to environmental hazards [13,25,27,61,62]. Beyond biological conditions and individual lifestyle and behavioral factors (such as diet and physical activity), other determinants are also related to childhood overweight or obesity, such as structural determinants (e.g., policies on childhood, health, food, and advertising) and environmental factors (either physical, such as ambient air pollution, or socioeconomic) [16,53]; for example, children’s food-related environment. Some studies show that low-income park neighborhoods are more likely to have a higher density of unhealthy establishments (e.g., fast-food stores and restaurants) compared to parks in high-income areas [66]. Others have found that there is a higher presence of unhealthy retailers around schools located in deprived areas [67]. 

Schools are also relevant environments to prevent overweight and obesity, through healthy diets at school and the promotion of physical activity (e.g., during breaktime, physical education classes, active transport from and to school, and extracurricular activities at school) [68]. In this respect, children—and those in Spain in particular—may practice more outdoor physical exercise in school facilities than after school, since most lunch breaks are between 2 and 2.5 h long, and many schools include sports programming during lunch. 

Barcelona’s unique context and urban planning are also important for understanding the divergence in the results of our study, which differs substantially from most of the scholarship, which is focused on cities from Anglo-Saxon countries. Since the 1990s, the city council has invested substantially to address green space and park space inequities through the municipality. Several neighborhoods have subsequently benefited from the Catalan 2004 Neighborhood Law (Llei de Barris) and the Barcelona 2016 Pla de Barris. These are aimed at addressing infrastructural needs in working-class neighborhoods, including those related to play and green space infrastructure. The 2012 Green and Biodiversity Plan also increased access to green space in 54 spaces throughout the city, while adding new urban connectors and corridors. As a result, Barcelona does not suffer from the types of green inequalities experienced by many cities in Europe or around the world [69,70,71,72], where one can find large differences in green and play access by neighborhood and by SES. In our study, children living in Barcelona in the least and the most deprived areas (i.e., first and fifth quintiles, respectively) have a similar quantity of overall and green play spaces near home.

Finally, this study took place in one of the densest cities in Europe, which, together with good availability and diversity of play spaces, may mean that most of the children living in Barcelona have a play space area within a walkable distance (300 m) from home. Moreover, our research only considered the city of Barcelona itself, which is located within a much larger metropolitan area, where the relation between residential proximity to play spaces and children overweight and obesity might be different. Many families from Barcelona might also use accessible green spaces just outside the city limits. For example, Barcelona is bordered by the large regional park of Collserola, an urban forest with several play spaces, which has become a growing source of leisure and play for families. All these unique characteristics might contribute to some of our findings and some of the inconsistencies we identified. 

### 4.3. Strengths, Limitations, and Future Research

To our knowledge, this is one of the first studies to evaluate the impact of residential proximity to play spaces on the development of childhood overweight or obesity in a dense city in Southern Europe. Indeed, our work is novel in exploring different types of urban play spaces, including the measure of play space diversity, in relation to childhood overweight or obesity, and to assess the impact of socioeconomic determinants moderating these associations in Southern Europe. The main strengths of our study are its longitudinal design, its long follow-up period (2011–2018), its objectively measured exposure and health outcomes, and the large sample size (more than 70,000 children). 

However, the research also has some limitations. Although we used an exposure variable that included the diversity of play spaces, we were not able to consider those features of the play spaces that determine their attractiveness and quality, or the children’s use and access; we also did not consider the size (area) of the play spaces, or different buffer distances and other proximities to play spaces (e.g., from school) [39,61]. Despite our longitudinal study design, the information on play space exposure was based on ecological variables from 2014. We used data from this year because it was the last available from the municipality and marked the midpoint (2014) of the period of analysis (2011–2018). Our measures of proximity to play spaces were based on census tract calculations, rather than individual exposure, which might introduce an information bias. This was also the case for socioeconomic status, which was estimated from the census tract deprivation level in 2011. The deprivation index was based on data from the 2011 census, which is a different year from that for play space exposure (2014), because it was the most recently available information. However, it has been shown that area SES does not suffer great variations between years and censuses [73]. Our study only included data from public healthcare centers, which might underrepresent children using private healthcare services, and especially those of higher-income families. In order to avoid non-differential overweight or obesity misclassification bias when calculating the hazard ratios, all subjects with overweight or obesity at baseline were excluded from our cohort to solve problems of using incidence risk ratios in follow-up studies [44]. 

Therefore, future studies should aim at collecting information on play space quality, access, infrastructure, use, and size, to explore their relation with overweight and obesity and their interaction with deprivation level [23,61]. Moreover, further analyses might consider using other buffer distances (e.g., 500 m or 1 km) and other residential proximities to play spaces to assess exposure. Indeed, how different spatial configurations (e.g., one large park versus several smaller green spaces) promote healthy communities and their variations across context, population groups, and urban designs is poorly understood [61,74]. Ethnographic qualitative studies would help to identify these relationships [13]. Furthermore, research including individual- or family-level data on SES would limit residual confounding in future studies [61], although in our sensitivity analyses, maternal BMI and maternal nationality, which are proxies of individual SES variables, did not confound the observed associations. Additionally, it is important to continue analyzing the role that exposure levels may have in the relationships between play spaces and overweight and obesity, as there may be certain thresholds and plateaus. Further studies in the context of Southern European countries, such as Spain, would help to understand the link between play infrastructures and childhood overweight and obesity. Thus, better evidence related to play spaces and their impact on health would be useful for local governments when taking actions to create healthy and child-friendly cities.

## 5. Conclusions

This study shows that residential proximity to play spaces is, in general, not associated with childhood overweight or obesity in Barcelona. Although it is not possible to draw clear conclusions from our study, we found some indications on the modifying role of area SES and level of residential exposure to play spaces. Access, quality, size, and other play space characteristics need to be assessed in further studies to continue understanding the effect of play spaces on children’s overweight or obesity. Based on the existing evidence, we call for cities to continue increasing and improving urban play spaces with an equity perspective.

## Figures and Tables

**Table 1 ijerph-19-13676-t001:** Descriptive characteristics of the study population (N = 75,608).

	Total Population	Remained Normal Weight during Follow-Up	Developed Overweight or Obesity	
	N = 75,608	N = 53,393 (70.62%)	N = 22,215 (29.38%)	*p*-Value *
Girls, N (%)	36,833 (48.72%)	26,523 (49.68%)	10,310 (46.41%)	<0.01
Age baseline, years median (p25; p75)	2.12 (2.06; 2.66)	2.12 (2.06; 2.65)	2.11 (2.06; 2.66)	<0.01
Age at case, years median (p25; p75)	6.32 (6.04; 8.24)	-	6.32 (6.04; 8.24)	-
Time of follow-up, years median (p25; p75)	5.77 (3.92; 8.61)	6.69 (4.23; 9.81)	4.10 (3.33; 5.94)	<0.01
Children’s nationality, N (%)				
Global North	70,638 (93.42%)	49,992 (93.63%)	20,646 (92.94%)	
Global South	4970 (6.57%)	3401 (6.37%)	1569 (7.06%)	<0.01
Area deprivation level (quintiles), N (%)				
First (least deprived)	15,124 (20.00%)	11,540 (21.61%)	3584 (16.13%)	
Second	15,118 (20.00%)	11,047 (20.69%)	4071 (18.33%)	
Third	15,126 (20.00%)	10,659 (19.96%)	4467 (20.11%)	
Fourth	15,120 (20.00%)	10,227 (19.15%)	4893 (22.03%)	
Fifth (most deprived)	15,120 (20.00%)	9920 (18.58%)	5200 (23.41%)	<0.01
Residential proximity to play spaces, median (p25; p75)				
Overall	12.00 (8.00; 17.00)	12.00 (8.00; 17.00)	12.00 (8.00; 17.00)	0.36
Green	6.00 (4.00; 9.00)	6.00 (4.00; 9.00)	6.00 (4.00; 9.00)	0.25
Diversity	0.93 (0.72; 1.01)	0.93 (0.72; 1.01)	0.93 (0.74; 1.01)	0.05

Abbreviations: p25 = 25th percentile, p75 = 75th percentile. For continuous variables, values are median and p25; p75. For categorical variables, absolute number and percentage. Note: Children included in our study were those who were normal weight at study entry and had at least two BMI measurements. Area deprivation level was assigned using the Índice de privación de la Sociedad Española de Epidemiología (IP2011). Quintiles were created based on the distribution of our study population. The proportion of children who developed only obesity between 2011–2018 was 12.6% (10,592 children). * Chi-square test for categorical variables, Student’s test for parametric distributions, and Mann–Whitney’s U test for non-parametric distributions.

**Table 2 ijerph-19-13676-t002:** Descriptive characteristics of the study population by area SES (quintiles of deprivation) (N = 75,608).

	Deprivation Index (Quintiles)					
	First (Least Deprived)	Second	Third	Fourth	Fifth (Most Deprived)	
	N = 15,124	N = 15,118	N = 15,126	N = 15,120	N = 15,120	*p*-Value *
Overweight or obesity, N (%)	3584 (23.70%)	4071 (26.93%)	4467 (29.53%)	4893 (32.36%)	5200 (34.39%)	<0.01
Girls, N (%)	7359 (48.66%)	7388 (48.87%)	7393 (48.88%)	7358 (48.66%)	7335 (48.51%)	0.97
Age baseline, years median (p25; p75)	2.13 (2.06; 3.00)	2.11 (2.06; 2.60)	2.11 (2.06; 2.59)	2.11 (2.05; 2.57)	2.11 (2.06; 2.58)	<0.01
Age at case, years median (p25; p75)	6.35 (6.05; 8.29)	6.32 (6.04; 8.24)	6.33 (6.05; 8.26)	6.35 (6.03; 8.24)	6.29 (6.04; 8.21)	0.07
Time of follow-up, years median (p25; p75)	5.86 (3.94; 8.77)	5.71 (3.92; 8.69)	5.67 (3.90; 8.56)	5.76 (3.92; 8.58)	5.60 (3.93; 8.45)	<0.01
Children’s nationality, N (%)						
Global North	14,671 (97.00%)	14,505 (95.94%)	14,188 (93.80%)	13,896 (91.90%)	13,378 (88.48%)	
Global South	453 (3.00%)	613 (4.06%)	938 (6.20%)	1224 (8.10%)	1742 (11.52%)	<0.01
Residential proximity to play spaces median (p25; p75)						
Overall	11.00 (8.00; 16.00)	13.00 (9.00; 18.00)	13.00 (9.00; 18.00)	12.00 (9.00; 17.00)	11.00 (7.00; 16.00)	<0.01
Green	6.00 (4.00; 8.00)	6.00 (4.00; 9.00)	7.00 (4.00; 10.00)	6.00 (4.00; 9.00)	6.00 (4.00; 9.00)	<0.01
Diversity	0.94 (0.69; 1.03)	0.92 (0.69; 1.01)	0.92 (0.76; 1.00)	0.93 (0.80; 1.01)	0.93 (0.69; 1.01)	<0.01

Abbreviations: p25 = 25th percentile, p75 = 75th percentile. For continuous variables, values are median and p25; p75. For categorical variables, absolute number and percentage. * Chi-square test for categorical variables, Student’s test for parametric distributions, and Mann–Whitney’s U test and Kruskal–Wallis’s test for non-parametric distributions.

**Table 3 ijerph-19-13676-t003:** Adjusted associations between residential proximity to play spaces (continuous and in tertiles) and the development of childhood overweight or obesity among all children and stratified by sex (N = 75,608).

		Play Spaces Residential Proximity			
		Continuous Exposure (per IQR) ^a^ HR (95% CI)	Categorical (Tertiles) Exposure Models		
			Low Exposure (T1), HR (95% CI) ^b^	Moderate Exposure (T2), HR (95% CI) ^b^	High Exposure (T3), HR (95% CI) ^b^
All	Overall play spaces	1.02 (1.00; 1.03)	(Ref.)	0.99 (0.96; 1.02)	1.02 (0.99; 1.05)
	Green play spaces	1.01 (1.00; 1.03)	(Ref.)	1.00 (0.97; 1.03)	1.03 (0.99; 1.06)
	Diversity play spaces	1.02 (1.00; 1.04)	(Ref.)	1.04 (1.01; 1.07)	1.03 (0.99; 1.07)
Boys	Overall play spaces	1.02 (0.99; 1.04)	(Ref.)	1.00 (0.96; 1.05)	1.02 (0.98; 1.07)
	Green play spaces	1.02 (1.00; 1.04)	(Ref.)	0.99 (0.95; 1.04)	1.04 (0.99; 1.09)
	Diversity play spaces	1.01 (0.98; 1.04)	(Ref.)	1.04 (0.99; 1.08)	1.03 (0.98; 1.07)
Girls	Overall play spaces	1.02 (0.99; 1.04)	(Ref.)	0.98 (0.93; 1.03)	1.01 (0.96; 1.06)
	Green play spaces	1.01 (0.99; 1.03)	(Ref.)	1.00 (0.96; 1.05)	1.02 (0.97; 1.07)
	Diversity play spaces	1.03 (1.00; 1.06)	(Ref.)	1.04 (0.99; 1.09)	1.04 (0.99; 1.09)

Abbreviations: T1: first tertile outdoor play spaces; T2: second tertile outdoor play spaces; T3: third tertile outdoor play spaces. Note: Models were adjusted by child’s age at baseline (categorical) in the strata statement, population aged 2–14 living in the children’s residential area, child’s sex (only for the non-stratified model “All”), child nationality and area deprivation index. ^a^ Hazard ratio (HR) reported by one IQR increase in outdoor play space indicators (overall play spaces per 9, green play spaces per 5, diversity play spaces 0.29) within 300 m from children’s census tracts of residence. Analyses are conducted separately for each (3) residential proximity to outdoor play space indicators. HR for variables of adjustment coincided for each play space indicator. ^b^ Hazard ratio (HR) reported by each tertile increase of outdoor play space indicators. Overall play spaces: T1 < 9, T2 9–15, T3 ≥ 15; Green play spaces: T1 < 5, T2 5–8, T3 ≥ 8; Diversity play spaces: T1 < 0.85, T2 0.85–0.98, T3 ≥ 0.98.

**Table 4 ijerph-19-13676-t004:** Adjusted associations between residential proximity to play spaces (continuous and in tertiles) and development of childhood overweight or obesity by area SES (quintiles of deprivation) (N = 75,608).

		Play Spaces Residential Proximity			
	Area Deprivation Level (Quintiles)	Continuous Exposure (per IQR) HR (95% CI) ^a^	Categorical (Tertiles) Exposure Models		
			Low Exposure (T1), HR (95% CI) ^b^	Moderate Exposure (T2), HR (95% CI) ^b^	High Exposure (T3), HR (95% CI) ^b^
Overall play spaces	First (least deprived)	1.03 (0.98; 1.08)	(Ref.)	0.94 (0.87; 1.02)	1.05 (0.97; 1.14)
	Second	1.06 (1.02; 1.11)	(Ref.)	1.04 (0.96; 1.12)	1.09 (1.01; 1.18)
	Third	1.00 (0.97; 1.04)	(Ref.)	0.91 (0.84; 0.98)	1.00 (0.92; 1.07)
	Fourth	0.97 (0.94; 1.01)	(Ref.)	1.00 (0.93; 1.07)	0.96 (0.89; 1.02)
	Fifth (most deprived)	1.02 (0.99; 1.05)	(Ref.)	1.05 (0.98; 1.12)	1.03 (0.96; 1.10)
Green play spaces	First (least deprived)	1.04 (0.99; 1.09)	(Ref.)	0.91 (0.84; 0.98)	1.07 (0.99; 1.16)
	Second	1.03 (1.00; 1.07)	(Ref.)	0.93 (0.85; 1.00)	1.01 (0.94; 1.09)
	Third	1.01 (0.98; 1.05)	(Ref.)	1.04 (0.97; 1.12)	1.05 (0.98; 1.12)
	Fourth	0.99 (0.96; 1.03)	(Ref.)	1.00 (0.93; 1.08)	1.02 (0.95; 1.09)
	Fifth (most deprived)	1.01 (0.98; 1.03)	(Ref.)	1.02 (0.99; 1.14)	1.00 (0.95; 1.10)
Diversity play spaces	First (least deprived)	0.99 (0.95; 1.05)	(Ref.)	1.03 (0.95; 1.13)	1.02 (0.95; 1.10)
	Second	1.03 (0.98; 1.07)	(Ref.)	1.05 (0.97; 1.13)	1.07 (0.99; 1.15)
	Third	1.02 (0.98; 1.06)	(Ref.)	1.10 (1.02; 1.18)	1.05 (0.97; 1.13)
	Fourth	1.03 (0.99; 1.08)	(Ref.)	1.01 (0.95; 1.09)	1.06 (0.99; 1.14)
	Fifth (most deprived)	1.01 (0.97; 1.06)	(Ref.)	1.01 (0.95; 1.08)	0.97 (0.91; 1.04)

Abbreviations: T1: first tertile outdoor play spaces (low exposure); T2: second tertile outdoor play spaces (moderate exposure); T3: third tertile outdoor play spaces (high exposure). Note: Models are adjusted by child’s age at baseline (categorical) in the strata statement, population aged 2–14 living in the children’s residential area, child’s sex, and child’s nationality. Analyses are conducted separately for each (3) residential proximity to outdoor play space indicators. HR for variables of adjustment coincided for each play space indicator. ^a^ Hazard ratio (HR) reported by one IQR increase in outdoor play space indicators (overall play spaces per 9, green play spaces per 5, diversity play spaces 0.29) within 300 m from children’s census tracts of residence in each quintile of deprivation index. ^b^ Hazard ratio (HR) reported by each tertile increase of outdoor play space indicators in each quintile of deprivation index. Overall play spaces: T1 < 9, T2 9–15, T3 ≥ 15; Green play spaces: T1 < 5, T2 5–8, T3 ≥ 8; Diversity play spaces: T1 < 0.85, T2 0.85–0.98, T3 ≥ 0.98.

## Data Availability

The authors do not have permission to share the datasets analyzed in this study. The data are not publicly available due to ethical and privacy issues.

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
