# Peer review of "Residential Proximity to Urban Play Spaces and Childhood Overweight and Obesity in Barcelona, Spain: A Population-Based Longitudinal Study"

_ijerph, 2022, doi:10.3390/ijerph192013676_

Round 1

Reviewer 1 Report

I appreciate the opportunity to review this manuscript. I believe that it is very important to carry out this type of research in the European and Spanish context in order to have robust evidence on this topic.

Next, I suggest some recommendations to improve the text.

Introduction

Lines 47 to 49: Spain is actually one of the European countries with the highest ow/ob dates (Rito, A.I.; Buoncristiano, M.; Spinelli, A.; Salanave, B.; Kunešová, M.; Hejgaard, T.; García Solano, M.; FijaÅ‚kowska, A.; Sturua, L.; Hyska, J.; et al. Association between Characteristics at Birth, Breastfeeding and Obesity in 22 Countries: The WHO European Childhood Obesity Surveillance Initiative—COSI 2015/2017. Obes. Facts. 2019, 12, 226–243). As it is described it seems that it is not a problem or that it has improved over the years, but the reduction has been practically insignificant. It really is still a high weight. Moreover, according to the evolution of the data from the National Health Survey, it can be seen that there has been an increase. (Ministerio de Sanidad, Servicios Sociales e Igualdad. Encuesta Nacional de Salud 2017; Ministerio de Sanidad, Servicios Sociales e Igualdad: Madrid, Spain, 2017. Available online: https://www.mscbs.gob.es/estadEstudios/estadisticas/encuestaNacional/ encuesta2017.htm)

I would emphasize in this sentence that the figures for ow/ob in the Spanish child population are high.

Lines 71-74: where?

Lines 74-77. This seems like a goal, I would put it at the end of the intro or this could be highlighted in the discussion emphasizing that it has been done in this way unlike other studies.

Line 101-102: "In this regard, our study explores the role of multiple play spaces exposure indicators (overall, green and diversity of them) in relation to overweight and obesity."

This sentence seems like it would fit better in results or discussion, as it is stating something before talking about the objective of the study.

Lines 103-104: "In total, we include more than 1,650 plays spaces and different types of play spaces (e.g., playgrounds, plazas, recreational sports fields, parks, gardens and urban forests)." 

This sentence would be part of the results section. It is also included in methodology (lines 145-146).

Line 105-108: Let it be clearer that what is specified in this sentence is the objective of the study.

Method:

Lines 132-134: “All included individuals were followed until they reached 15 years old, became overweight or obese, transferred-out of SIDIAP, died or until the end of the study period (December 31, 2018).”

This sentence is not very clear. how can individuals who are no longer within SIDIAP or who have died have been followed? Please redo for clarification.

Line 169: "children were categorized as overweight or normal weight."

As a suggestion for improvement, I would change the word overweight to excess weight or overweight/obesity, if by this we mean excess weight including overweight and obesity for clarity. Also, I would mention it at the same time. Another option would be to use overweight/obesity to refer to this. In this way, it would not be necessary to clarify it repeatedly as it is done in this paragraph.

There are too many databases from different years. For the 2014 parks, for the 2011 SES level... It should be better explained or justified.

Results:

In general: Separate or differentiate with smaller font and with a space between the text and the table the text at the end of these. 

Line 274: I would change the title of section 3.2. Main results, to a title that represents what is included in that section.

Discussion:

Line 403: a period is missing after citing reference 23.

464-470: In my opinion it is not relevant to include this paragraph in relation to what is being discussed.

In the discussion it is commented that the quality and accessibility of green and play areas could facilitate their use. It can be included as future lines of research.

Highlight the need to carry out studies in the Spanish context in order to have solid evidence.

Reviewer 2 Report

The manuscript "Residential Proximity to Urban Play Spaces and Childhood Overweight and Obesity in Barcelona, Spain: A Population-Based Longitudinal Study", this study presents a robust methodology. It adequately justifies the negative results in the discussion, and provides new data regarding this interesting topic, and opens new perspectives for future studies regarding urban play spaces.

Author Response

We would like to thank the editor for the kind words and appreciate our work.

Reviewer 3 Report

This is a well-written study investigating the prospective association between the residential proximity to play spaces and unhealthy weight status in childhood. I have a few comments for this study as follows.

Abstract

- Please mention the sample size in the abstract. 

Methods

- In a sub-section of the design and study population, please add a figure or flow diagram illustrating how the final sample size was achieved.

- The authors might also consider using other buffer distances, such as 500 m or 1 km as additional robustness analyses. 

- Please elaborate more on how the time of follow-up was defined.

- The authors might consider using multilevel survival analyses since some variables were collected at different levels (individual vs. neighbourhood levels)

- Why were other confounders (e.g., parental education, income, etc.) not included?

Results

- As previously suggested, the authors might need to use different (or longer) buffer distances to understand the association between residential proximity to play spaces and child weight. A longer distance can capture cumulative opportunities for contact with nearby play spaces or green spaces. 
